# The Phenotype and Secretory Activity of Adipose-Derived Mesenchymal Stem Cells (ASCs) of Patients with Rheumatic Diseases

**DOI:** 10.3390/cells8121659

**Published:** 2019-12-17

**Authors:** Ewa Kuca-Warnawin, Urszula Skalska, Iwona Janicka, Urszula Musiałowicz, Krzysztof Bonek, Piotr Głuszko, Piotr Szczęsny, Marzena Olesińska, Ewa Kontny

**Affiliations:** 1Department of Pathophysiology and Immunology, National Institute of Geriatrics, Rheumatology and Rehabilitation, 02-637 Warsaw, Poland; ula.skalska@gmail.com (U.S.); zaklad.patofizjologii@spartanska.pl (I.J.); zbiu@op.pl (U.M.); ewa.kontny@spartanska.pl (E.K.); 2Department of Rheumatology, National Institute of Geriatrics, Rheumatology and Rehabilitation, 02-637 Warsaw, Poland; krzysztof.bonek@gmail.com (K.B.); piotr.gluszko@spartanska.pl (P.G.); 3Clinic of Connective Tissue Diseases, National Institute of Geriatrics, Rheumatology and Rehabilitation, 02-637 Warsaw, Poland; piotr_szczesny@live.com (P.S.); marzena.olesinska@vp.pl (M.O.)

**Keywords:** adipose-derived mesenchymal stem cells, phenotype, secretory potential, ankylosing spondylitis, systemic lupus erythematosus, systemic sclerosis

## Abstract

Mesenchymal stem/stromal cells (MSCs) have immunosuppressive and regenerative properties. Adipose tissue is an alternative source of MSCs, named adipose-derived mesenchymal stem cells (ASCs). Because the biology of ASCs in rheumatic diseases (RD) is poorly understood, we performed a basic characterization of RD/ASCs. The phenotype and expression of adhesion molecules (intracellular adhesion molecule (ICAM)-1 and vascular cell adhesion molecule (VCAM)-1) on commercially available healthy donors (HD), ASC lines (n = 5) and on ASCs isolated from patients with systemic lupus erythematosus (SLE, n = 16), systemic sclerosis (SSc, n = 17) and ankylosing spondylitis (AS, n = 16) were analyzed by flow cytometry. The secretion of immunomodulatory factors by untreated and cytokine-treated ASCs was measured by ELISA. RD/ASCs have reduced basal levels of CD90 and ICAM-1 expression, correlated with interleukin (IL)-6 and transforming growth factor (TGF)-β1 release, respectively. Compared with HD/ASCs, untreated and tumour necrosis factor (TNF) + interferon (IFN)-γ (TI)-treated RD/ASCs produced similar amounts of prostaglandin E2 (PGE_2_), IL-6, leukemia inhibiting factor (LIF), and TGF-β1, more IL-1Ra, soluble human leukocyte antigen G (sHLA-G) and tumor necrosis factor-inducible gene (TSG)-6, but less kynurenines and galectin-3. Basal secretion of galectin-3 was inversely correlated with the patient’s erythrocyte sedimentation rate (ESR) value. IFN-α and IL-23 slightly raised galectin-3 release from SLE/ASCs and AS/ASCs, respectively. TGF-β1 up-regulated PGE_2_ secretion by SSc/ASCs. In conclusion, RD/ASCs are characterized by low basal levels of CD90 and ICAM-1 expression, upregulated secretion of IL-1Ra, TSG-6 and sHLA-G, but impaired release of kynurenines and galectin-3. These abnormalities may modify biological activities of RD/ASCs.

## 1. Introduction

Rheumatic diseases (RD), triggered by a complex interplay of genetic and environmental factors and mediated by autoimmune and/or autoinflammatory mechanisms, are characterized by chronic inflammation, progressive damage and functional impairment of affected tissues and organs [1]. Systemic lupus erythematosus (SLE) is a multisystemic disease of autoimmune background. Overactivation of B cells, production of numerous autoantibodies, and defective immunoregulation are pivotal in SLE pathogenesis [2]. Systemic sclerosis (SSc), a rare autoimmune disease, is characterized by vascular derangement, abnormal fibroblast activation and progressive multi-organ fibrosis [3]. In SSc and SLE, serious life-threatening manifestations are common. Patients require long-term conventional immunosuppression but, despite an improvement in treatment opportunities, both diseases are still incurable [2,3]. Ankylosing spondylitis (AS) is accompanied by inflammatory back pain, damage to joint structures, and pathological bone formation, leading to spine ankylosis. Early long-lasting anti-inflammatory therapy may slow the progression of this irreversible structural damage [4]. This disease has certain autoinflammatory features and is mediated by interactions between innate and adaptive immune cells and cytokines [4,5].

Mesenchymal stromal/stem cells (MSCs), present in many embryonic and adult tissues, are endowed with regenerative potential and exert immunomodulatory effects on different components of the immune system, acting through cell-to-cell contact and/or secreted factors [6,7]. For these reasons, MSCs are thought to represent a promising therapeutic tool in autoimmune and inflammatory diseases [8]. However, depending on local conditions, tissue-resident MSCs can exhibit either anti- or pro-inflammatory capabilities, and thus may exert either protective effects or contribute to disease development [9]. In SLE and SSc, a growing body of data indicates numerous abnormalities of bone marrow derived MSCs (BM-MSCs), suggesting their possible contribution to disease pathology and raising questions about their autologous therapeutic application [10,11,12,13]. The knowledge about BM-MSC biology in AS is much less; limited data show aberrant function of these cells, and therefore several ongoing clinical trials apply allogenic MSCs [14,15,16]. Because of low expression of class II major histocompatibility complex (MHC) molecules, MSCs are claimed to be immune privileged and allogenic MSC therapy is generally regarded as harmless [17]. However, animal studies question this dogma by showing recipient immune responses and rejection of transplanted allogenic MSCs [18,19]. In humans, the clinical consequences of the therapeutic application of MHC-mismatched MSCs are unknown and, if possible, administration of autologous MSCs is recommended [19]. Human adipose tissue is a rich source of MSCs that possess stronger immunomodulatory capability than BM-MSCs [20,21]. Little is known about the biological properties of adipose tissue-derived MSCs (ASCs) from SLE, SSc and AS patients. These cells have been assessed in several SLE and SSc animal models with promising results [10]. At present, several clinical trials for curing SLE and SSc patients with allogeneic or autologous stromal vascular fraction (SVF), obtained from adipose tissue, are in progress (National Clinical Trials: NCT 02975960, NCT 02866552, NCT 02741362, clinicaltrials.gov) [22]. Unfortunately, SVF preparations contain variable proportions of ASCs and other cells, which may lead to unpredictable effects, makes standardization difficult and limits SVF therapeutic application [10]. By contrast, ASCs represent a more homogeneous population of cells and early phase clinical trials reported beneficial effects of autologous ASC transplantation in the alleviation of cutaneous symptoms in SSc patients [23,24,25]. However, there is still controversy over the use of autologous or allogeneic ASCs in the clinic because there are studies suggesting functional alterations of these cells in SSc [26,27]. Therefore, with the view of the potential therapeutic application of autologous ASCs in mind, we have performed a basic characterization of these cells and compared the phenotype and secretory potential of RD patients’ ASCs (RD/ASCs) with the corresponding features of ASC lines originating from healthy donors (HDs).

## 2. Materials and Methods

### 2.1. Patients and Sample Collection

Three groups of patients who fulfilled the criteria for SLE (n = 16), SSc (n = 17) or AS (n = 16), were included in the study [28,29,30]. This study meets all criteria contained in the Declaration of Helsinki and was approved by the Ethics Committee of the National Institute of Geriatrics, Rheumatology, and Rehabilitation, Warsaw, Poland (approval protocol number: KBT-8/4/20016). All patients gave their written informed consent prior to enrolment.

### 2.2. ASC Isolation and Culture

Specimens of subcutaneous abdominal fat (approximately 300 mg) were taken from the patients by 18 G needle biopsy. Tissue processing, ASC isolation and culture were performed as described previously [31]. Five human adipose-derived mesenchymal cell lines (Lonza Group, Lonza Walkershille Inc., MD, USA, donor numbers: 0000440549, 0000410252, 0000535975, 0000605220, 0000550179) were used as a control. All experiments were performed using ASCs at 3–5 passages. The medium used for ASC culture was composed of DMEM/F12 (PAN Biotech UK Ltd., Wimborn, UK), 10% fetal calf serum (FCS) (Biochrom, Berlin, Germany), 200 U/mL penicillin, 200 µg/mL streptomycin (Polfa Tarchomin S.A., Warsaw, Poland) and 5 µg/mL plasmocin (InvivoGen, San Diego, CA, USA). For some experiments, ASCs were stimulated for 24 h with recombinant human tumor necrosis factor (TNF)-α and interferon (IFN)-γ (both from R & D Systems, Minneapolis, MN, USA; each used at 10 ng/mL), 2000 U/mL of IFN-α, (R&D Systems), 5 ng/mL of transforming growth factor (TGF-β1) or 20 ng/mL of interleukin (IL)-23 (both from PeproTech Inc. Rocky Hill, NY, USA). Then, culture supernatants (SNs) and cells were harvested for further analysis by enzyme-linked immunosorbent assays (ELISAs) or flow cytometry, respectively.

### 2.3. Flow Cytometry Analysis

For ASC phenotype analysis, cells were treated with non-enzymatic cell dissociation solution (ATCC Manassas, VA, USA) and washed with FACS buffer (phosphate-buffered saline, 0.1% NaN3, 1% FCS). Then, 5 × 10^4^ cells were resuspended in 50 μL of FACS buffer and stained for 30 min on ice with fluorochrome conjugated murine anti-human monoclonal antibodies against the following surface markers: CD90-FITC, CD105-PE, CD73-APC (eBioscience, San Diego, CA, USA), CD34-PE-Cy7, CD45-PE, CD19-PE and CD14-APC (BD Pharmingen, San Diego, CA, USA). For evaluation of intracellular adhesion molecule 1 (ICAM-1) and vascular cell adhesion molecule 1 (VCAM-1) expression, ASCs were detached as described above, then stained with anti-ICAM-1-PE and anti-VCAM-1-APC antibodies (both from Biolegend, San Diego, CA, USA). After the washing step, cells were acquired and analyzed using a FACSCanto cell sorter/cytometer and Diva software. The gating strategy and applied isotype controls are shown in Appendix A.

### 2.4. ELISAs

The concentrations of cytokines were measured in culture SNs in duplicate using specific ELISAs. The IL-6 ELISA was performed according to our own procedure using a goat polyclonal neutralizing antibody specific for human IL-6 (R & D Systems, Minneapolis, MN, USA), and an IL-6 specific rabbit polyclonal antibody (Sigma-Aldrich, St. Louis, MO, USA) as the capture and detection antibodies, respectively, followed by horseradish peroxidase-conjugated goat anti-rabbit immunoglobulins and o-phenylenediamine dihydrochloride (OPD) (both from Sigma) as a substrate. Human recombinant IL-6 (R & D Systems) was used a standard. The concentrations of TGF-β1, galectin-3, leukemia inhibiting factor (LIF) and IL-1 receptor antagonist (IL-1Ra) were measured using ELISA DuoSet kits, while PGE_2_ was measured using the Parameter kit (all from R & D Systems). The soluble form of human leukocyte antigen G (sHLA-G) concentration was measured using a specific ELISA kit from Biovendor, Brno, Czech Republic. Tumor necrosis factor-inducible gene 6 protein (TSG-6) concentration was assessed using a specific ELISA from RayBiotech, Norcross, GA, USA. Kynurenine concentration was measured spectrophotometrically as described elsewhere [32]. Briefly, SNs were mixed with 30% trichloroacetic acid at a 2:1 ratio and incubated for 30 min at 5 °C, then centrifuged at 10,000× *g* for 5 min and finally diluted at a 1:1 ratio in Ehrlich’s reagent (100 mg p-dimethyl benzaldehyde and 5 mL glacial acetic acid; Sigma-Aldrich). The optical density of the samples was measured at wavelength of 490 nm. L-kynurenine (Sigma-Aldrich) diluted in culture medium was used to prepare the standard curve.

### 2.5. Data Analysis

Data were analyzed using GraphPad Prism software version 7. The Shapiro–Wilk test was used as a normality test. The results are shown as median ± interquartile range (IQR) or range. One-way analysis of variance (ANOVA) with repeated measures and post-hoc Tukey test was used to compare untreated and cytokine-treated ASCs. The differences between ASC lines from healthy donors (HD/ASCs) and ASCs from SLE (SLE/ASCs), SSc (SSc/ASCs) and AS (AS/ASCs) patients were analyzed using the Kruskal–Wallis and Dunn’s multiple comparison tests. For comparison of two groups (e.g., HD/ASCs vs. SLE/ASCs for IFN-α-treated cells), the Mann–Whitney U test was applied. Parametric (Pearson’s linear) and non-parametric (Spearman’s rank) correlation tests were used to assess an association between tested parameters. Probability values less than 0.05 were considered significant.

## 3. Results

### 3.1. Patients

The patient cohort was heterogeneous with respect to demographic and clinical data (Table 1). There were no significant differences between patient groups in body mass index (BMI), disease duration, and erythrocyte sedimentation rate (ESR) values, but SLE patients were younger than SSc patients, and AS patients had a slightly higher concentrations of C-reactive protein (CRP) than other patients. All AS patients were HLA-B27 positive and they were mostly treated with non-steroid anti-inflammatory drugs (NSAIDs). The majority of SLE and SSc patients had disease-specific autoantibodies (anti-dsDNA or Scl70, respectively) and received immunosuppressive drugs, usually with (SLE) or without (SSc) glucocorticosteroids. A similar proportion of SSc patients had localized (52.9%) or diffused (47%) disease form. A minority of patients received non-biologic disease-modifying anti-rheumatic drugs (DMARDs).

### 3.2. Phenotype of ASCs

Almost all HD/ASCs and RD/ASCs possessed MSC specific surface markers (CD105, CD90, CD73) and the percentage of triple positive cells was similar in every group (Figure 1A). There was also no significant difference in the level of CD105 and CD73 marker expression, shown as the median fluorescence intensity (MFI). However, RD/ASCs expressed less CD90 molecules than HD/ASCs, both on triple (CD105+/CD90+/CD73+) (Figure 1B) and single (CD90+) (data not shown) positive cells. The proportion of ASCs expressing hematological markers was low and similar in every group, i.e., less than 4% of HD/ASCs and RD/ASCs were positive for CD14, CD19, CD45, or CD34 (Figure 1C,D). However, four RD/ASC lines contained ≥10% CD34+ cells. The median percentage of HLA-DR^+^ cells was below 10, and the majority of HD/ASCs and RD/ASCs lines contained less than 1–2% of HLA-DR+ cells. Nevertheless, some of them (one HD/ASC and seven RD/ASCs) contained ≥20% of these cells (Figure 1D). There was a strong inverse correlation between CD90 MFI on HD/ASCs and basal secretion of PGE_2_ by these cells (Figure 1E), while in the RD group, CD90 MFI correlated positively but moderately with IL-6 secretion (Figure 1F). Both HD/ASCs and RD/ASCs differentiated in vitro into osteoblastic, chondrogenic and adipogenic lineages (data not shown; manuscript in preparation).

### 3.3. Expression of Adhesion Molecules by ASCs

A similar proportion of untreated and cytokine (TNFα + IFNγ; TI) treated HD/ASCs and RD/ASCs expressed the surface adhesion molecules ICAM-1 and VCAM-1 (Figure 2A,B). VCAM-1 was co-expressed by a small fraction of ICAM-1+ cells and no single VCAM-1+ cells were found. By contrast, the frequency of single ICAM-1+ cells predominated over the proportion of double ICAM-1+/VCAM-1+ cells. Amongst untreated cells, a slightly higher proportion of SSc/ASCs co-expressed ICAM-1 and VCAM-1 compared to other RD/ASCs (Figure 2A). Upon TI treatment, the percentage of single positive, double positive, and consequently all ICAM-1+ cells increased significantly (Figure 2B). Basal levels (MFI) of VCAM-1 expression on HD/ASCs and RD/ASCs were similar (Figure 2C), and TI treatment did not modify it (Figure 2D). However, basal levels of ICAM-1 expression on single and double positive cells were significantly lower in SSc/ASCs and AS/ASCs than in HD/ASCs, and in the case of SLE/ASCs, a similar tendency was also noted (Figure 2C). Upon TI treatment, these differences disappeared (Figure 2D). In the RD group, the ICAM-1 MFI on ASCs positively correlated with TGF-β secretion by these cells (Appendix A).

### 3.4. Basal and Cytokine-Triggered Secretory Activity of ASCs

As shown in Figure 3A,B, both untreated and cytokine-stimulated RD/ASCs secreted significantly smaller amounts of kynurenines and galectin-3 than HD/ASCs. Treatment with TI, but not other cytokines, significantly increased kynurenine secretion in HD/ASCs and, to a lesser extent, in RD/ASC cultures (Figure 3A), but failed to change the release of galectin-3 by both HD/ASCs and RD/ASCs (Figure 3B). However, the slight elevation of galectin-3 secretion by RD/ASCs was observed when disease-specific cytokines (IFN-α for SLE/ASCs and IL-23 for AS/ASCs) were used as the stimuli (Figure 3B). In addition, basal release of galectin-3 by RD/ASCs correlated weakly and inversely with patient ESR value, especially in the SSc group in which the correlation was much stronger (Figure 3C,D). On the other hand, RD/ASCs released more IL-1Ra, sHLA-G and TSG-6 than HD/ASCs, and cytokine treatment had a significant effect on their secretion (Figure 4). By contrast, both untreated and cytokine-stimulated HD/ASCs and RD/ASCs released similar amounts of the other tested factors (PGE_2_, IL-6, TGF-β and LIF) (Figure 5). Upon TI treatment, HD/ASCs and RD/ASCs upregulated their secretion of IL-6; there was also increased release of LIF from HD/ASCs and PGE_2_ from RD/ASCs while no changes were observed in the case of TGF-β, with the only exception of a smaller secretion of this cytokine by TI treated compared to untreated SSc/ASCs.

## 4. Discussion

MSCs are not constitutively immunosuppressive but acquired such capabilities upon exposure to appropriate local environments, and control immune responses through cell–cell contact and paracrine secretion of numerous soluble factors [9]. To assess the basic biological features of RD/ASCs, the cells exposed in vivo to chronic inflammatory milieu, we first analyzed the phenotype of these cells. According to a revised statement of the International Fat Applied Technology Society (IFATS), there are three minimal criteria for the definition of ASCs: plastic adherence, differentiation potential into adipocytes, chondrocytes and osteoblasts, as well as characteristic surface phenotype. More than 80% of these cells should express primary stable positive markers (CD105, CD90, CD73), while the expression of primary negative markers (CD45 and others) should be below 2% [33]. Other authors observed expression of positive and negative markers on more than 90% and less than 5% of ASCs, respectively [34]. The present results show that ASCs isolated from subcutaneous abdominal adipose tissue of RD patients fulfill the above phenotypic criteria of MSCs (Figure 1A,C,D). Observed individual variation in the proportion of CD34+ cells in ASC lines of both HD and RD patients (Figure 1D) may be explained by unique features of this molecule, which is a primary unstable ASC positive marker, present at variable levels and lost gradually during cell culture [33]. Although MSCs are thought to be devoid of surface HLA-DR, expression of these molecules on ASCs has been reported [35]. Thus, presence of HLA-DR molecules on some HD/ASC and RD/ASC lines, observed in our study (Figure 1D), is unsurprising. Altogether, our results show that in terms of surface marker expression, RD/ASCs resemble ASCs obtained from healthy volunteers, which is consistent with other reports [36,37,38].

In comparison with HD/ASCs, we found significantly reduced levels of CD90 expression on RD/ASCs (Figure 1B). CD90 is a glycoprotein endowed with numerous immunological and non-immunological functions, including cell–cell and cell–matrix interactions, cell motility, inflammation, and fibrosis [39]. It was reported that CD90 expressed on MSCs promotes osteogenic differentiation, but inhibits adipogenic differentiation of these cells, while MSCs with reduced expression of CD90 lose their immunosuppressive activity [40,41]. Therefore, it is likely that impaired CD90 expression on RD/ASCs contributes to the abnormal regenerative and immunoregulatory function of these cells. Consistent with this supposition, our results suggest that the level of CD90 expression may affect the secretory potential of ASCs (Figure 1E,F). According to our observation, high CD90 expression may restrict PGE_2_ release in HD/ASCs, while it may promote IL-6 secretion in RD/ASCs. The reason for CD90 downregulation on RD/ASCs is unknown, but prolonged in vivo exposure of these cells to inflammatory milieu is the probable cause. Unfortunately, we failed to notice any significant changes in CD90 expression after three days exposure of HD/ASCs and RD/ASCs to TI stimulation (data not shown). Although this observation does not rule out the likelihood of CD90 loss due to the action of other factors contributing to chronic inflammation, alternative explanations, e.g., intrinsic defects, also exist [12].

The adhesion molecules ICAM-1 and VCAM-1 regulate the migration of MSCs and provide their interaction with co-operating immune cells, making them indispensable for MSC-mediated immunosuppression [42,43]. Similar to other reports [18], we failed to observe single VCAM-1 bearing ASCs, found mostly single positive ICAM-1 cells and, less frequently, cells co-expressing both of these molecules (Figure 2A). Importantly, compared with HD/ASCs, both single and double positive RD/ASCs had lower basal ICAM-1 expression (Figure 1C), which, in addition, correlated moderately with TGF-β1 release by these cells (Appendix A), suggesting that ICAM-1 may also have an impact on RD/ASC secretory function. Upon TI treatment, there was a similar upregulation of ICAM-1, but not VCAM-1, expression on HD/ASCs and RD/ASCs (Figure 1B,D). Therefore, the reason for impaired basal ICAM-1 expression on RD/ASCs is not related to pro-inflammatory cytokine exposure in vivo. Based on the present findings, we conclude that reduced basal expression levels of CD90 and ICAM-1 are characteristic for RD/ASCs. Neither the reason nor the consequences of these abnormalities are known yet. We did not find any association between basal CD90 and ICAM-1 expression and patient demographic and clinical data, including age, BMI, disease duration, ESR/CRP values, disease activity, etc. (data not shown). Nevertheless, our results suggest that both defects may affect RD/ASC secretory activity.

MSCs regulate immune responses by the release of various immunomodulatory factors and indoleamine 2,3-dioxogenase (IDO)-1-mediated catabolism of tryptophan into kynurenines [6,35,44]. Our results show significant differences between RD/ASCs and HD/ASCs in the secretion of these factors. Firstly, both untreated and cytokine-treated RD/ASCs produced less kynurenines and galectin-3 (Figure 3A,B). Activation of the kynurenine pathway, triggered mostly by IFN-γ, mediates the immunosuppressive action of MSCs in vitro and in vivo and exerts strong inhibitory effects mostly on the proliferation and cytotoxicity of T cells and natural killer (NK) cells [44]. Galectin-3, a member of the β-galactoside binding protein family, is a multifunctional protein with immunomodulatory effects on cell apoptosis, activation, differentiation and migration [45]. This protein is a critical mediator of MSC suppression of T cell proliferation and its secretion is considered as a biomarker for the immunomodulatory potential of MSCs [46]. We observed that TI treatment upregulated kynurenine release by HD/ASCs and RD/ASCs, but the latter cells responded poorly, suggesting that they are rather refractory to kynurenine pathway activation. In addition, we found that disease-specific cytokines were unable to trigger the kynurenine pathway in ASCs, while IFN-α and IL-23 activated, to a small extent, the galectin-3 pathway in SLE/ASCs and AS/ASCs, respectively. In contrast, secretion of galectin-3 by HD/ASCs was stable, which is consistent with its reported constitutive expression in BM-MSCs [47]. Since we found an inverse correlation between basal secretion of galectin-3 by RD/ASCs and ESR value (Figure 3C,D), it is likely that impairment of galectin-3 production by ASCs may contribute to inefficient control of systemic inflammation in these patients.

Secondly, compared with HD/ASCs, we found significant elevation of IL-1Ra, sHLA-G, and TSG-6 secretion by RD/ASCs (Figure 4). Applied stimuli did not alter the release of these factors. IL-1Ra counteracts the inflammatory effects of IL-1; its role in mediating the immunosuppressive effect of MSCs lies mostly in the polarization of macrophages toward the anti-inflammatory M2 phenotype and inhibition of B cell differentiation [48]. In collagen-induced arthritis in mice, inhibition of T-helper (Th)17 cells generation by IL-1Ra producing MSCs was also shown [49]. HLA-G, a non-classical HLA class I molecule, inhibits functions of innate and adaptive immune cells. Both surface-expressed and soluble forms (present in body fluids) of HLA-G are involved in the suppression of the cytotoxic function of T cells and NK cells, the maturation and function of dendritic cells (DC), and the generation of regulatory T cells (Tregs) [50]. TSG-6 mediates many of the immunomodulatory and reparative activities of MSCs and exerts therapeutic effects in various animal disease models [50]. This protein acts by binding to numerous ligands, e.g., extracellular matrix components and chemokines, where it inhibits neutrophil migration and macrophage activation and promotes M2 macrophage switching and Treg generation [51].

Independently of activation status, HD/ASCs and RD/ASCs secreted similar amounts of the other tested factors (PGE_2_, IL-6, TGF-β1, and LIF; Figure 5). In general, RD/ASCs and HD/ASCs responded in similar way to applied stimuli, and a significant increase of secretion, if any, was triggered by TI treatment. The only exception was an elevation of PGE_2_ release by TGF-β1 treated SSc/ASCs (Figure 5). All these ASC-derived factors were reported to mediate the immunomodulatory effects of MSCs in several ways, including inhibition of T cell proliferation. In addition, IL-6 and PGE_2_ were shown to inhibit maturation of DCs and mediate switching of M1 inflammatory macrophages to an M2-like phenotype, while LIF and TGF-β1 were suggested to contribute to Treg generation [52,53,54].

As discussed above, there is a redundancy in the biological activities of MSC secreted factors. In addition, to achieve therapeutic immunosuppressive effects, a combination of these factors is usually required. For example, in SLE patients, the therapeutic effect of allogeneic MSC application was accompanied by an upregulation of Tregs and downregulation of Th17 cells, which was dependent on sHLA-G and TGF-β1, or PGE_2_ action, respectively [55,56]. Therefore, it is hard to predict how the presently found abnormalities in the secretory activity of RD/ASCs may affect the immunoregulatory function of these cells. Nevertheless, observed upregulation of IL-1Ra, sHLA-G and TSG-6 secretion and normal release of IL-6, PGE_2_, TGF-β1 and LIF suggest that RD/ASCs, exposed in vivo to a chronic inflammatory microenvironment, may preferentially use these factors to control excessive immune responses. However, impaired kynurenine and galectin-3 secretion may contribute to inefficiency of ASC control. It is likely that galectin-3 deficiency is more critical, as low production of this protein by RD/ASCs is associated with higher-grade systemic inflammation. Interestingly, disease-specific cytokines seem to trigger a compensatory mechanism by upregulating the release of galectin-3 from ASCs in SLE and AS and PGE_2_ secretion in SSc patients.

Our study has some limitations. Firstly, it is well known that the type of harvesting procedure used may affect viability, yield or the biology of ASCs [57]. Because of ethical reasons, we had to apply the needle-aspiration technique, which is not commonly used. However, for ASC isolation, we utilized the enzymatic digestion-based method [31], intended mainly for experimental purposes [57]. In addition, to perform these analyses, isolated ASCs had to be expanded in vitro. Fortunately, the applied ASC isolation method turned out to be good enough to obtain a homogeneous population of cells with the MSC phenotype (Figure 1), even from scanty specimens of adipose tissue. Secondly, our RD patient cohort was heterogeneous in terms of some demographic data (age), clinical symptoms and treatment (Table 1). Obviously, these differences may also influence the phenotype and secretory activity of ASCs. Surprisingly, despite all the differences between SLE, SSc and AS patients, their ASCs show similar alterations in CD90 and ICAM-1 expression and immunoregulatory factor release. Because the aforementioned RDs are inflammatory disorders, chronic in vivo exposure of ASCs to the inflammatory environment of adipose tissue is a more likely explanation for the observed abnormalities.

## 5. Conclusions

In summary, we report that RD/ASCs are characterized by a low basal level of CD90 and ICAM-1 expression, elevated spontaneous secretion of IL-1Ra, TSG-6 and sHLA-G, but impaired release of kynurenines and galectin-3, and there are only small differences between ASCs obtained from SLE, SSc, and AS patients. Exposure of RD/ASCs to a pro-inflammatory cytokine cocktail (TI) normalizes ICAM-1 expression, but not the secretion of immunoregulatory factors. We suggest that the observed alterations are caused by in vivo exposure of ASCs to an inflammatory milieu and may contribute to the inadequate immunoregulatory function of these cells.

## Figures and Tables

**Figure 1 cells-08-01659-f001:**
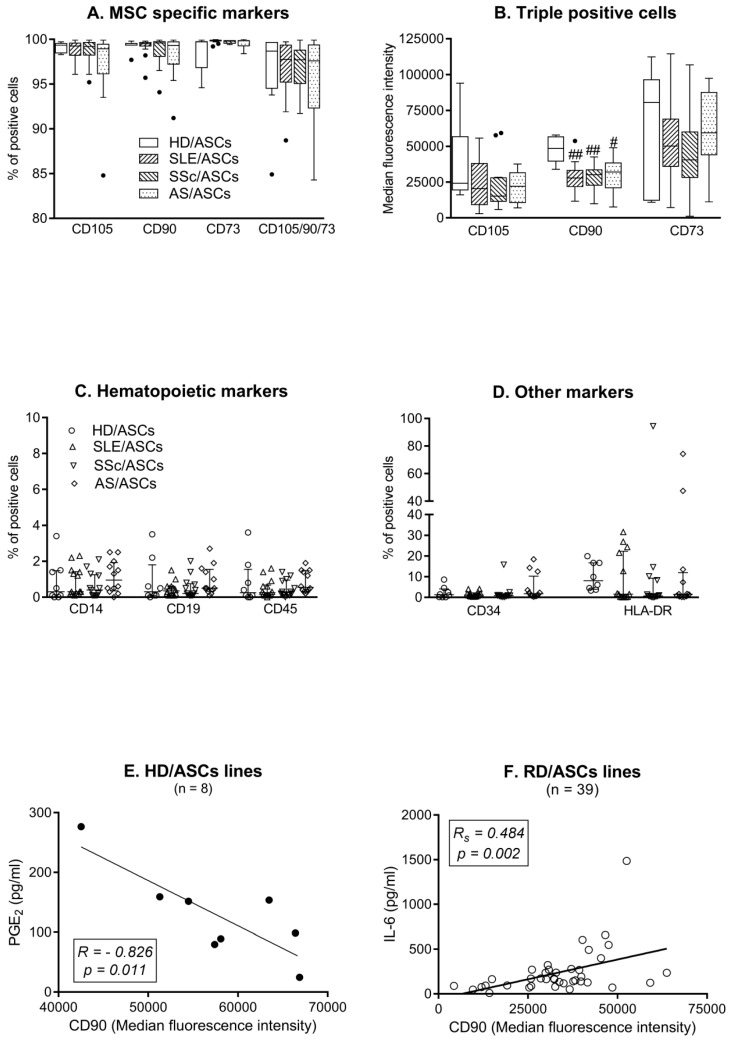
The phenotype of ASCs from healthy donors (HD) and patients with rheumatic diseases (RD). Expression of MSC specific (**A**,**B**) and non-specific (**C**,**D**) markers was assessed quantitatively (**B**) and/or qualitatively (**A**,**C**,**D**) using ASCs from healthy donors (HD/ASCs; n = 5), systemic lupus erythematosus (SLE/ASCs; n = 14), systemic sclerosis (SSc/ASCs; n = 13) and ankylosing spondylitis (AS/ASCs; n = 12) patients. Data are expressed as the median (horizontal line) with interquartile range (IQR, box), lower and upper whiskers (data within 3/2xIQR) and outliers (points) (Tukey’s box) (**A**,**B**) or as the median with IQR (**C**,**D**). ^#^
*p* = 0.05–0.01; ^##^
*p* = 0.01–0.001 for HD/ASCs versus RD/ASCs comparison. Pearson’s (*R*) and Spearman’s rank (*R_s_*) correlation coefficients. Solid cirle—HD/ASC samples, hollow circles RD/ASC samples (**E**,**F**).

**Figure 2 cells-08-01659-f002:**
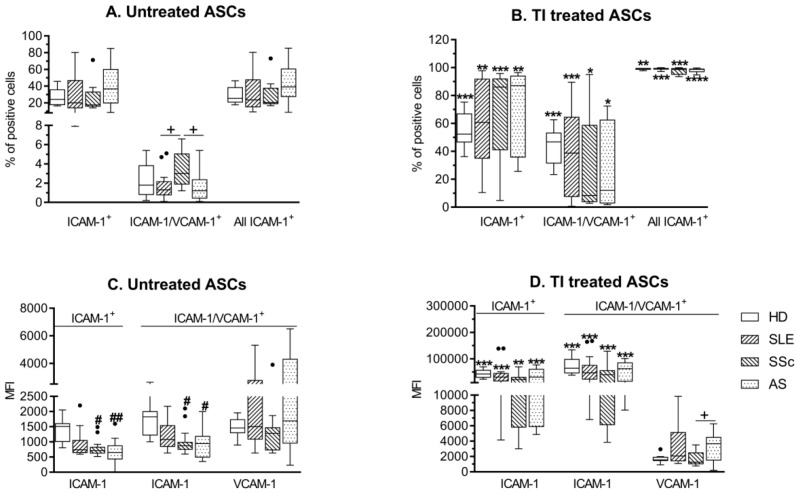
Expression of adhesion molecules on untreated and cytokine-treated ASCs. Healthy donors (HD; n = 5), systemic lupus erythematosus (SLE; n = 16), systemic sclerosis (SSc; n = 16) and ankylosing spondylitis (AS; n = 15) ASCs were cultured in medium alone (untreated ASCs) or in the presence of tumor necrosis factor (TNF)-α and interferon (IFN)-γ (TI treated ASCs) for 24 h. The qualitative (**A**,**B**) and quantitative (**C**,**D**) assessments of intracellular adhesion molecule (ICAM)-1 and vascular cell adhesion molecule (VCAM)-1 expression were performed. Results are expressed as the Tukey’s boxes (see Figure 1). ^#^
*p* = 0.05–0.01, ^##^
*p* = 0.01–0.001 for HD/ASCs versus patients’ ASCs, while * *p* = 0.05–0.01, ** *p* = 0.01–0.001, *** *p* = 0.001–0.0001 for untreated versus TI treated ASC comparisons, + *p* = 0.05–0.01 for patients’ versus patients’ ASCs. Solid circles (points) represents outliers.

**Figure 3 cells-08-01659-f003:**
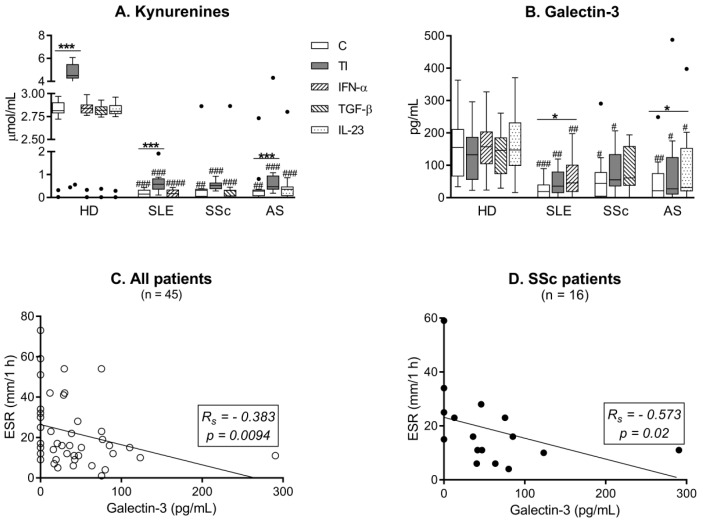
Impaired secretion of galectin-3 and kynurenines by ASCs of patients with rheumatic diseases (RD). Five healthy donor (HD) ASCs lines from two different passages (n = 10), systemic lupus erythematosus (SLE; n = 16), systemic sclerosis (SSc; n = 16–17) and ankylosing spondylitis (AS; n = 15) patients’ ASCs were cultured in medium alone (C, control), or in the presence of the indicated cytokines (TI, TNF-α + IFN-γ). Galectin-3 and kynurenine concentrations were measured in culture supernatants. (**A**,**B**) Data are expressed as Tukey’s boxes (see Figure 1); * *p* = 0.05–0.01, *** *p* = 0.001–0.0001 for control versus cytokine-treated cells; ^#^
*p* = 0.05–0.01, ^##^
*p* = 0.01–0.001, ^###^
*p* = 0.001–0.0001, ^####^
*p* < 0.0001 for HD/ASCs versus RD/ASCs comparisons. (**C**,**D**) *R_s_*, Spearman’s rank correlation coefficient. Hollow circles represents RD/ASC samples, solid circle represents SSc/ASC samples. ESR - erythrocyte sedimentation rate.

**Figure 4 cells-08-01659-f004:**
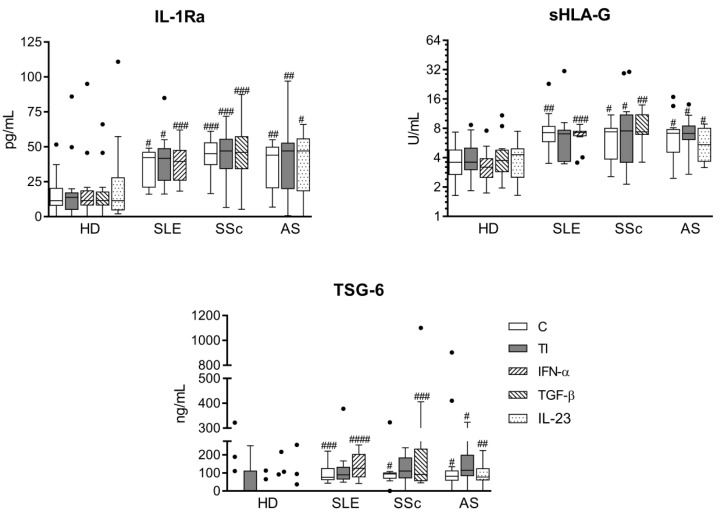
Enhanced secretion of IL-1Ra, soluble human leukocyte antigen G (sHLA-G) and tumor necrosis factor-inducible gene (TSG)-6 by ASCs of patients with rheumatic diseases (RD). Culture conditions of ASC lines from healthy donors (HD; n = 10), systemic lupus erythematosus (SLE; n = 12–16), systemic sclerosis (SSc; n = 10–17) and ankylosing spondylitis (AS; n = 9–15) patients were the same as described in Figure 3. Concentrations of indicated factors were measured in culture supernatants. Data are expressed as the Tukey’s boxes (see Figure 1). No differences between untreated (C) and cytokine-stimulated cells were found. ^#^
*p* = 0.05–0.01; ^##^
*p* = 0.01–0.001; ^###^
*p* = 0.001–0.0001; ^####^
*p* < 0.0001 for HD/ASCs versus RD/ASCs comparison.

**Figure 5 cells-08-01659-f005:**
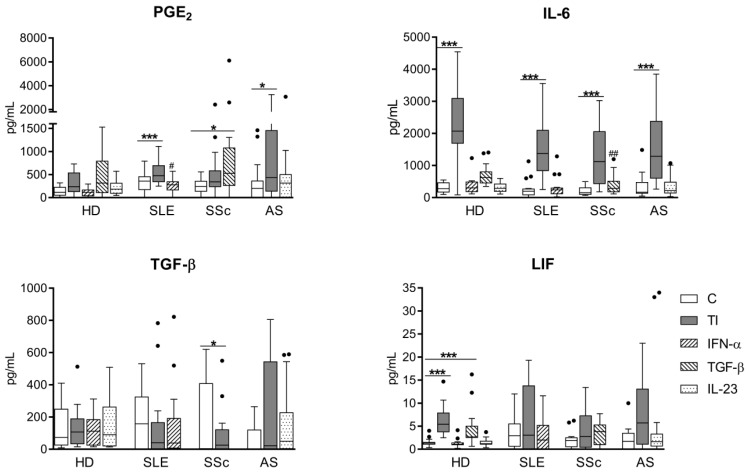
Similar secretion of other factors by ASCs of healthy donors and patients with rheumatic diseases. Culture conditions of ASC lines from healthy donors (HD; n = 10) and from patients with rheumatic diseases (RD), i.e., systemic lupus erythematosus (SLE; n = 16), systemic sclerosis (SSc; n = 16–17) and ankylosing spondylitis (AS; n = 15), were the same as described in Figure 3. Concentrations of indicated factors were measured in culture supernatants. Data are expressed as the Tukey’s boxes (see Figure 1). * *p* = 0.05–0.01, *** *p* = 0.001–0.0001 for untreated (C) versus cytokine-treated cells; ^#^
*p* = 0.05–0.01, ^##^
*p* = 0.01–0.001 for HD/ASCs versus RD/ASCs.

**Table 1 cells-08-01659-t001:** Demographic and clinical characteristics of the patients.

Parameters	Systemic Lupus Erythematosus (SLE) (n = 16)	Systemic Sclerosis (SSc) (n = 17)	Ankylosing Spondylitis (AS) (n = 16)
Demographics			
Age, years	41 (20–54) ^#^	52 (20–77)	43 (25–70)
Sex, female (F)/male (M), n	15 F/1 M	12 F/7 M	8 F/8 M
BMI	24.3 (16.4–39.1)	25.8 (16.5–38.7)	26.9 (21.4–35.8)
Disease duration, years	8 (0–47)	3 (1–23) ^a^/5 (1–40) ^b^	6 (1.5–17)
Clinical data			
Disease activity *, score	7 (0–32)	1 (0–8)	6.3 (1.0–8.2)
Laboratory values			
CRP, mg/L	5 (1–23)	5 (3–18)	8 (5–59) ^##^
ESR, mm/h	16.5 (3–73)	16 (4–59)	15 (1–59)
Proteinuria, mg/24 h	185 (0–7550)	0 (0–0.2)	n/a
C3, mg/dL	73.5 (23.2–133)	98.1 (65.8–141)	n/a
C4, mg/dL	15.45 (5.38–20.6)	17.35 (13–27.1)	n/a
ANA, titre (1:x)	960 (160–10,240)	2560 (320–20,480)	n/a
anti-dsDNA antibody, %	75	n/a	n/a
anti-dsDNA antibody, IU/mL	68.85 (0–666.9)	n/a	n/a
Scl-70 antibody, %	n/a	88.9	n/a
Autoantibody specificities, no.	3 (1–7)	3 (2–4)	n/a
Medications, %			
NSAIDs			81.25
Immunosuppressive drugs	92.8	55	0
Non-biologic DMARDs	28.6	27.3	37.5
Glucocorticosteroids	75	23.5	21.25

Except where indicated otherwise, values are the median (range). BMI, body mass index; * SLEDAI, SLE Disease Activity Index, * EUSTAR, the European League Against Rheumatism Scleroderma Trials and Research revised index, or * BASDAI, Bath Ankylosing Spondylitis Disease Activity Index; CRP, C-reactive protein; ESR, erythrocyte sedimentation rate; C, complement components; ANA, antinuclear antibody; Scl-70, anti-topoisomerase I antibody; NSAIDs, non-steroid anti-inflammatory drugs; DMARDs, disease-modifying anti-rheumatic drugs; n/a, not applicable. ^#^
*p* = 0.03 for SLE vs. SSc patient comparison; ^##^
*p* = 0.03 for AS vs. SLE and SSc comparisons.

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
