# Peer review of "The Phenotype and Secretory Activity of Adipose-Derived Mesenchymal Stem Cells (ASCs) of Patients with Rheumatic Diseases"

_cells, 2019, doi:10.3390/cells8121659_

Round 1

Reviewer 1 Report

This is a valuable contribution to the literature on the expression of various biomarkers by rheumatic diseases/adipose-derived stem cells (RD/ASC) and the comparison of those biomarkers from patients with SLE, systemic sclerosis and ankylosing spondylitis with a few commercially available stem cell lines. The main take-home message of this paper was the low basal level of CD90 and ICAM-1 expression and elevated secretion of IL-1Ra, TSG-6 and sHLA-G, but importantly, impaired release of kynurenines and galactin-3 by RD/ASCs. 

Main Concern:

The authors are encouraged to address the impact of treatment of SLE, SSc and AS patients with immunosuppressive drugs and their impact on the results of this study. This point was not covered in any depth in the Discussion. It is also likely to be the case that various biomarkers of disease activity listed in Table 1 were influenced by drug therapy. For example, the consequence of therapy of SLE patients with corticosteroids on T-cell activity has been previously addressed and led to a fundamental change in methodology including the need to capture T-cells from peripheral blood of patients diagnosed with SLE, BEFORE, any therapy with corticosteroids was implemented.

Author Response

Response to Reviewer 1

We would like to thank the Reviewer for comments and valuable suggestion concerning the impact of RD patients therapy on ASCs biology. We have taken this question into account in the last, presently added, paragraph of the Discussion. Obviously, it is possible that all of used drugs (NSAIDS, DMARDs, glucocorticosteroids and immunosuppressive drugs) may modify ASCs physiology. Our patients’ cohort was heterogeneous with respect to demographic and clinical data and we expected to find some disease specific features of ASCs obtained from patients suffering from different RD. However, contrary to expectations, ASCs from SLE, SSc and AS patients show similar abnormalities. These patients were treated with different drugs, but have one feature in common – chronic systemic inflammation. Therefore, we supposed that chronic exposure of ASCs in vivo to inflammatory milieu is a more likely explanation of observed alterations than the type of applied therapy.

All changes made in the revised version of the manuscript are traced in the “Marked Copy” of the manuscript.

Reviewer 2 Report

Dear Editor, 

I send you my review comments on the article “The phenotype and secretory activity of adipose-derived mesenchymal stem cells (ASCs) of patients with rheumatic diseases” of Ewa Kuca-Warnawin et al. 

This experimental work evaluates the phenotype and secretory activity of adipose stem cells derived from rheumatic diseases, comparing them with those from healthy donors. In particular, the Authors performed an immunophenotypic characterization for mainly stemness markers and adhesion molecules.

The manuscript title entirely represents data presented, and overall the topic is quite of interest.

Abstract section: Authors reported several acronyms, but they do not immediately write the full name. I suggest to specify them. Probably an abbreviation’s section should be inserted.  Introduction section: Is known that rheumatic diseases are chronic, multisystem disorders characterized by a markedly inflammatory environment, and some of them are chronic systemic inflammatory disease (such as rheumatoid arthritis). In this scenario it is very important assayed some inflammatory factors, like PGE2 and IL-6. To better describe the pathological state of these diseases, I suggest describing in detail the reason why PGE2 and IL-6 have been measured.  Line 92, 2.2 “ASCs isolation and culture” paragraph: Authors report this sentence “Specimens of subcutaneous abdominal fat were taken from the patients by 18 G needle biopsy.” If the adipose tissue was collected after a needle biopsy, I suppose that the sample is similar to a lipoaspirate sample. Is it true? How many volumes of the sample was collected from each patient?  Line 169, “3.2 Phenotype of ASCs”: Authors should report the full name of HD (Healthy Donors). Figure 1: The graphs A, C, and D in figure 1 show the percentage of positive cells and several positive and negative stemness markers. Why did the graph B report the median fluorescence intensity? Please justify or make the graph similar to others. Also, why the Authors use the name “Triple positive ASCs” for adipose stem cells? Probably this name should add to cells after the immunophenotypic characterization. Line 246 and 372, In the legend of figure 4 and at the end of the “conclusions” are reported two dots. Please, correct it. Discussion section: In my opinion, Discussion is very long, I suggest making it shorter. Moreover, discussion mainly recaps the results but does not discuss the findings in depth. The authors have characterized the RD/ASCs in term of stemness markers expression and secretory activity, however, they don't argue anything about the potential mechanisms involved in this disease. Authors could more extensively speculate about the clinical aspects. Nowadays are there clinical trials about ASCs in rheumatic diseases? Some concepts presented in the discussion could be moved to the introduction. Also, the Authors did not cite studies about the autologous use of Adipose derived-stem cells. Is known that the quality and quantity of ASCs strictly depend on the adipose tissue or lipoaspirate handling techniques, on patient’s variability, on purifying method and storage condition. The reference n.28 cited (line 271) do not argue about the “quality” of adipose-derived stem cells used. In order to improve the understanding of the text, and to deepen this important topic, I suggest to discuss these information including two bibliographic references, such as the work of Palumbo P et al. (Methods of Isolation, Characterization and Expansion of Human Adipose-Derived Stem Cells (ASCs): An Overview. Int J Mol Sci. 2018 Jun 28;19(7). pii: E1897. doi: 10.3390/ijms19071897) and the work of Chu DT et al. (Adipose Tissue Stem Cells for Therapy: An Update on the Progress of Isolation, Culture, Storage, and Clinical Application. J Clin Med. 2019 Jun 26;8(7). pii: E917. doi: 10.3390/jcm8070917.) The Authors demonstrated a difference in expressed markers in HD/ASC and RD/ASC. Could these differences be due to the use of anti-inflammatory drugs or glucocorticoids? Please, discuss it.

In conclusion, I consider that this manuscript can be suitable for publication in Cells

Kind regards

Author Response

Response to Reviewer 1

We would like to thank the Reviewer for comments and valuable suggestions which are addressed below.

Similarly to other selected factors, IL-6 and PGE2 secretion was measured because both factors are known to mediate immunomodulatory effects of MSCs. This is explained in the Discussion (lanes366-369). Of course, IL-6 and PGE2, produced mainly by monocytes/macrophages, contribute to RD pathogenesis as well, but this question is beyond the scope of this study.

Yes, samples of adipose tissue taken from RD patients resemble lipoaspirate. It is possible to get about 300 mg tissue by this method and to obtain ASCs number enough to perform described analyses, the cells have to be expanded in vitro (by about 3-4 weeks) – this information is added to the section 2.2. and to the end of the Discussion.   Explanation of HD (healthy donors) abbreviation is added to the last lane of Introduction and to the Abstract. In the Abstract TI abbreviation is also explained, other abbreviations are commonly used acronyms and their full names are shown in the section 2.4. According to your suggestion we added section ”Abbreviations” to the manuscript.

The number (%) of triple positive (CD105/CD90/CD73+) cells is shown on Fig. 1A, while on Fig. 1B the expression levels (MFI) of particular markers on these cells are displayed. The title of Fig. 1B is corrected.

Printing errors are corrected.

Discussion – Information about clinical trials using adipose tissue SVF and ASCs in SSc and SLE is added to the Introduction. Data concerning AS are missing yet. Regarding the question of the impact of harvesting procedure and isolation method on the “quality” and functionality of ASCs, suitable paragraph is added to the end of the Discussion. However, in our opinion there is no need to discuss this question comprehensively, because due to ethical reasons in RD patients there is no choice to apply method other than needle-biopsy. We do not agree that the Discussion should be changed and focused on speculation about the clinical aspects of our results. All particular observations are discussed and every paragraph ends with some suggestions concerning consequences of stated alteration to ASCs biology, especially immunomodulation. This suggestion is also added to the Conclusion. Based on present results it is hard to draw “firm conclusions”, but we are working on the evaluation of immunomodulatory functions of RD/ASCs and hope to get soon data that allow to discuss this question more deeply. Our suggestion about possible cause of observed ASCs alterations is added to the end of Discussion and to the Conclusion.

Reference 28 was replaced with more suitable one.

All changes are traced in the “Marked Copy” of the manuscript.

Kind Regards

Ewa Kuca-Warnawin